# Three-Way DNA Junction as an End Label for DNA in Atomic Force Microscopy Studies

**DOI:** 10.3390/ijms231911404

**Published:** 2022-09-27

**Authors:** Zhiqiang Sun, Tommy Stormberg, Shaun Filliaux, Yuri L. Lyubchenko

**Affiliations:** Department of Pharmaceutical Sciences, University of Nebraska Medical Center, Omaha, NE 68198, USA

**Keywords:** Atomic Force Microscopy (AFM), three-way junction (3WJ), DNA label, nucleosomes

## Abstract

Atomic Force Microscopy (AFM) is widely used for topographic imaging of DNA and protein-DNA complexes in ambient conditions with nanometer resolution. In AFM studies of protein-DNA complexes, identifying the protein’s location on the DNA substrate is one of the major goals. Such studies require distinguishing between the DNA ends, which can be accomplished by end-specific labeling of the DNA substrate. We selected as labels three-way DNA junctions (3WJ) assembled from synthetic DNA oligonucleotides with two arms of 39–40 bp each. The third arm has a three-nucleotide overhang, GCT, which is paired with the sticky end of the DNA substrate generated by the SapI enzyme. Ligation of the 3WJ results in the formation of a Y-type structure at the end of the linear DNA mole cule, which is routinely identified in the AFM images. The yield of labeling is 69%. The relative orientation of arms in the Y-end varies, such dynamics were directly visualized with time-lapse AFM studies using high-speed AFM (HS-AFM). This labeling approach was applied to the characterization of the nucleosome arrays assembled on different DNA templates. HS-AFM experiments revealed a high dynamic of nucleosomes resulting in a spontaneous unraveling followed by disassembly of nucleosomes.

## 1. Introduction

Atomic Force Microscopy (AFM) is one of the widely used nanotools for single molecule studies of protein-DNA complexes of various types [1]. In many such AFM studies, precise position determination of proteins is of utmost importance, which needs to be retrieved from these experiments. Bulky markers on DNA are necessary for unambiguous location of the protein, and end-labeling of DNA is a straightforward approach used for labeling. Streptavidin is one such marker, as it specifically binds biotin, which can be incorporated at the end of the DNA template [2,3,4]. The use of proteins as a marker has several complications. First, binding of the protein marker can require specific composition of the buffer that can complicate the assembly of the complex under study. Second, if the marker is not covalently bound, it can dissociate from the binding site. Third, bulky protein appearance can introduce ambiguity to analysis [5], a problem that led us to utilize rhizavidin, a smaller-sized streptavidin analog [6,7] to reliably distinguish nucleosomal particles due to the similar size of streptavidin to nucleosomes in the AFM images [8]. If the DNA template has a single-stranded end, single-stranded binding proteins such as SSB of *E. coli* can be an attractive marker [9,10], but nonspecific or unwanted interactions can occur [11]. We mentioned the buffer composition above, which can modulate both DNA and protein properties [12,13,14,15]. This factor is critical in studies of labile and dynamic systems such as chromatin [16,17,18,19]. These complications can be avoided if other types of bulky markers are used. An interesting approach was proposed in [20], where a single-stranded DNA (ssDNA) loop was used as a terminal label for AFM studies, but the yield of labeling was low. Later experiments increased the yield of the terminal ssDNA marker to ~70% [21]. However, the high flexibility of ssDNA and its relatively low contrast in AFM images compared with dsDNA is a complication in the identification of ssDNA loops at the end of DNA duplexes for some studies. Reliable visualization of a terminal label in HS-AFM studies, is particularly important, as multiple frames per second are used to describe the dynamics of a system. The use of a dsDNA-based label could help overcome these complications.

Here, we describe a labeling method based on the use of branched DNA constructs as bulky terminal labels for linear DNA molecules. We used a three-way junction (3WJ) as the simplest branched DNA molecule, which is covalently ligated to a sticky end of the DNA template. We describe a methodology for assembling an appropriate 3WJ, which is stable and easily identifiable with AFM. Our DNA label can be produced with high efficiency and serves as a reliable terminal label for DNA and protein-DNA complexes in both AFM experiments performed in air and time-lapse HS-AFM studies in an aqueous environment. We tested the use of this labeling in several different protein-DNA systems. This paper presents the results of AFM studies of mononucleosomes and dinucleosomes assembled on the linear DNA template labeled with the 3WJ.

## 2. Results

### 2.1. Coupling of Linear DNA with Three-Way DNA Junction

The 3WJ was assembled by annealing three synthetic DNA oligonucleotides (Figure 1A) to form a construct shown in Figure 1B. Oligonucleotide Y2 contains six T residues to provide additional flexibility to the joint of the junction [22,23]. The lengths of the arms of the junction are 40 bp, 39 bp, and 20 bp between Y1–Y2, Y2–Y3, and Y1–Y3, respectively. The 3WJ construct was extracted from the gel and ligated to the sticky end of the DNA with its 3 nt overhang. DNA-3WJ samples were then deposited on functionalized APS-mica for AFM imaging as described in the methods section. One scan of representative AFM images of the labeled DNA is shown in Figure 2A. White arrows indicate the presence of the 3WJ label on the substrate. The yield of DNA labeling was measured by counting the number of clearly defined DNA labels compared with the total of clearly defined DNA molecules and was determined to be 69% (n = 198), indicating successful visualization of the label using AFM.

AFM visualization revealed different conformations of the DNA label at the end of the substrate. Representative zoomed-in images of different label conformations are shown in Figure 2B. Three classes of DNA label conformations were identified: an open “T” conformation with two arms almost with 180° angle between the arms (shown in Figure 2B(i)), “Y” label conformation with an acute angle between the arms (shown in Figure 2B(ii)), and a condensed “bulge” label conformation (shown in Figure 2B(iii)). The probability of each label is shown in Table 1. The “T” conformation was found the be the most common (*p* = 0.53), followed by the “Y” conformation (*p* = 0.39). The “bulge” conformation was the minor species (*p* = 0.08). These data indicate the dynamics of the 3WJ described previously [24]. The Y-junction is flexible, allowing the arms to adopt different, but still clearly visible, conformations. The high yield of T-confirmations is due to the stacking of opposite DNA duplexes facilitated by the T-bulge in the Y2 arm [22,23].

To further verify the assembly of the construct and explore the flexibility of the 3WJ, we performed contour length measurements of the DNA substrate and the DNA label. A histogram of the DNA contour length measurements is shown in Figure 3. DNA measured from the free end to the center of the DNA label, presented as a histogram in Figure 3A, showed a narrow distribution with a peak Gaussian value of 142 ± 10 nm (S.D.), indicating a homogenous population of DNA substrates that were not perturbed by the terminal labeling process. The length distribution of the DNA label, measured from one end of the label to the other and shown as a histogram in Figure 3B, showed a broader distribution, with a peak Gaussian value of 33 ± 4 nm (S.D.). This is likely due to the flexible nature of the 3WJ label, as it can adopt different conformations and degrees of compaction.

### 2.2. HS-AFM to Visualize the Dynamics of the 3WJ Directly

We deposited our labeled DNA on functionalized mica and imaged it in our imaging buffer on our HS-AFM, capable of capturing multiple images per second, allowing for the observation of dynamics of the system. The data of the entire set of frames are assembled in Appendix A. Snapshots from the movie are shown in Figure 4. The label in the “T” conformation from the first frame is immediately observable. It is shown on the right terminus of the DNA molecule. The DNA is shown to be mobile, highlighting the ability of HS-AFM to observe the dynamics of complexes. The unlabeled DNA terminus shows high mobility, curving, and looping throughout the video. The label shows flexibility as well; while the “T” conformation is prominent for the first thirty frames of the video, it briefly adopts a “Y” conformation in frame 31 before reverting to a “T” in frame 33. The DNA and label move throughout the video, but, significantly, the label is visible throughout. In addition to direct visualization of the dynamics of 3WJ, these data illustrate the use of the 3WJ labeling for time-lapse experiments.

### 2.3. Assembly of Nucleosomes on Y-DNA Substrate

After verifying the successful assembly of our terminally labeled DNA substrate, we aimed to test whether this assembly could be used to study protein-DNA complexes. We chose nucleosomes as our protein-DNA complex of study. Nucleosomes were assembled on the nonspecific 356 bp DNA substrate terminally labeled with the 3WJ using the gradient dialysis approach described in the methods section. After assembly, complexes were diluted to 2 nM, deposited on functionalized mica, and imaged using AFM.

Representative AFM images of nucleosome assembly with selected snapshots are shown in Figure 5. The bright white features shown in the images are the nucleosome core particles, with DNA flanking either side. White arrows in Figure 5A indicate the 3WJ labels. Indeed, we are able to both successfully assemble nucleosome complexes on the labeled DNA substrate and visualize the 3WJ label. Snapshots in Figure 5B show several examples of successfully assembled and labeled nucleosome complexes. The position of the nucleosome varies throughout the different snapshots, as indicated by the distance of the core particle from the labeled end of the DNA. Using the 3WJ as a fiducial marker, one can accurately determine the DNA sequence occupied by the nucleosome based on the position relative to the 3WJ label.

Next, we sought to utilize the 3WJ label to characterize the role of the DNA sequence on the nucleosome assembly. To accomplish this goal, we constructed the DNA substrate comprising the 601 nucleosome-specific sequence along with a 372 bp segment with no specific affinity to the histone core termed random sequence. A diagram of the DNA construct can be seen in Appendix A and labeled with the 3WJ as described in the methods. This DNA construct has sufficient space for binding at least two nucleosomes. A 147 bp sequence 601-motif with a very high affinity to the nucleosome assembly was placed 80 bp from the opposite side as the 3WJ label. We assembled nucleosomes on the substrate and imaged the complexes. AFM images are shown in Figure 6, with selected snapshots to the right. These images show that mononucleosomes are assembled at the position distant to the label with the 98% preference, indicating a strong affinity of the nucleosome assembly on the 601-motif. This conclusion is supported by the mapping results shown in Figure 7. In this graph, the nucleosome positions (orange dots) are located at the 601-sequence, with a few nucleosomes bound to the non-specific DNA sequence. There was no binding of the nucleosomes to the 3WJ label. These results indicate that the 3WJ is a reliable terminal label, able to facilitate studies of sequence-dependent protein-DNA interactions by allowing the precise determination of the protein position.

### 2.4. Dynamics of the Nucleosome Arrays

We continued our HS-AFM experiments to characterize the dynamics of the nucleosome arrays. In one set of experiments, we followed the dynamic unraveling of a mononucleosome assembled on the labeled DNA substrate. The video is shown as a movie in Appendix A, and selected snapshots from this video are shown in Figure 8. In frame 1, we see a clear assembled nucleosome, shown as the bright globular feature indicated with an arrow. The DNA label, which is distant from the nucleosome, remains in the “T” conformation. Over time, the nucleosome undergoes unraveling, identified by decreasing the nucleosome height and increasing arm lengths. Unwrapping of DNA from the histone core starts after frame 10, with a complete dissociation in frame 25. This conclusion is supported by the height measurements of nucleosomes shown in Appendix A. The initial height of the nucleosome with the value of 2.5 nm gradually decreases over time to 1 nm (frame 17), after which the core dissociates. Note that the 3WJ label in this frame is still clearly visible with primarily T-shape briefly adopting a “Y” conformation as shown in frame 20.

In another set of experiments, we followed the unraveling process of a dinucleosome complex. The movie set is shown in Appendix A, and a few snapshots from this video are shown in Figure 9. This complex is identified as a dinucleosome by the presence of two bright circular features along the DNA substrate. The nucleosomes are separated with one nucleosome located close to the 3WJ, as shown in frame 14. This nucleosome unravels rapidly, so in frame 16, only one nucleosome remains. In this set, the 3WJ label is not immediately visible. The label reappears on the expected DNA terminus in frames 23 and 27, as the second nucleosome begins to unravel. The second nucleosome starts unraveling between frames 23 and 33 and is completely dissociated from the DNA in frame 39 (30).

## 3. Discussion

We presented here a methodology for the DNA end labeling with a bulky feature routinely identifiable with AFM. We selected the simplest branched DNA molecule, the 3WJ. This approach has a number of attractive features. First, the vast majority of topographic AFM studies are performed with protein molecules and their assemblies [4,25,26]. Topographically on AFM images, they appear as globular features. The branched morphology of the 3WJ label, being distinct from the globular features of proteins, allows for no ambiguity in identifying the label compared with the protein. We illustrated this benefit by imaging nucleosomes appearing as globular features on the linear DNA template. Second, the 3WJ label is covalently attached to the end of DNA, so buffer composition is not an issue for this type of labeling. This property is important for studies of protein complexes, as the stability of such complexes can depend dramatically on the buffer composition.

The 3WJ is a dynamic structure with preferable pyramidal conformation with 60° between the arms [22,23]. The unpaired bases at the joint point switch the equilibrium to the T-conformation of the paired arms, and we observed this type of conformational transition with AFM. Such conformational transition does not affect the visualization of the label, although T-conformation of the junction is simpler to visualize.

Time-lapse AFM imaging is highly attractive for biological studies, as it allows one to perform imaging close to physiological conditions by avoiding sample drying [27,28]. The use of HS-AFM instrumentation is especially attractive, as it will enable the direct visualization of dynamics at a high data acquisition rate [29,30,31]. We demonstrated here that labeling with 3WJ is fully compatible with such AFM instrumentation. The high yield of labeling is another attractive feature of the described approach that we would like to emphasize.

The 3WJ is visible through most of the frames, despite its dynamics. The arm can be off the surface becoming non-visible, but appears on the following frame allowing for unambiguous identification of the DNA end. Once we know which end is the 3WJ end, we can track it through each frame, even if in some frames the 3WJ is not visible on AFM.

The DNA template as prepared remains stable during the typical storage of DNA without requiring special conditions for the sample storage. The ligation of 3WJ to the DNA with sticky ends requires adjustment of the junction sequences to the DNA template sequence. However, incorporating the restriction site sequence into the primer for the PCR synthesis of the DNA template allows one to use the unmodified 3WJ with any DNA substrate desired, providing that the substrate does not already contain the SapI restriction site within. In the case where another restriction enzyme is required, the 3WJ sequence can be easily modified. This is another convenience, as assembled 3WJ can be used for labeling a nearly unlimited number of different DNA templates.

In conclusion, we successfully designed and implemented an efficient and clearly identifiable 3WJ DNA label. This label is observable in the majority of complexes imaged and was shown to be useful in both AFM and HS-AFM studies. Thus, the results presented in this section can be utilized in a vast array of AFM studies in the future, particularly those involving nucleosomes and other protein-DNA complexes.

## 4. Materials and Methods

### 4.1. Oligonucleotides for 3WJ Assembly

Oligonucleotides of the following sequences were acquired commercially from Integrated DNA Technologies (Coralville, IA, USA) with PAGE purification:

Y1: 5′ GCTATACAGCTCGCCGCAGCCGAACGCCCTTGCGCAGCGAGTCAGTGAGATAGGAAGCGGAAGAGCG-3′.

Y2: 5′-CGCTCTTCCGCTTCCTATCTCACTGACTCGCTGCGCAAGGTTTTTTCTAACAGCATCACACACATTAACAATTCTCACATCTGGG-3′.

Y3: 5′-CCCAGATGTGAGAATTGTTAATGTGTGTGATGCTGTTAGGCGTTCGGCTGCGGCGAGCTGTAT-3′.

### 4.2. Preparation of 3WJ End Label

The oligonucleotides comprising the 3WJ were designed to have complementarity to one another in order to form the “Y” shape of the 3WJ (Figure 1). A six-nucleotide thymine repeat was added to Y2 to facilitate assembly of the T structure of the Y-end [22,23]. A three-nucleotide (GCT) overhang was added to Y1 to facilitate ligation to the full DNA construct with a complementary sticky end generated with SapI restriction enzyme. The oligonucleotides used and a schematic of the 3WJ are shown in Figure 1. To form the 3WJ, the three oligonucleotides (Y1, Y2, Y3) were mixed at an equimolar ratio and annealed by heating to 95 °C. Assembly was verified by gel electrophoresis. A representative gel can be seen in Appendix A.

### 4.3. Preparation of Full DNA Construct

Two DNA substrates were used in the experiments. The first is a nonspecific DNA sequence 356 bp in length. The second is a 600 bp sequence capable of binding two nucleosomes. This sequence contains the strong nucleosome positioning 601 motif at the far end of the template, as seen in Appendix A [32]. The duplexes are generated from PCR using a plasmid vector pUC19 with these primers. The primer includes the cutting region for the restriction enzyme SapI (5′-GAAGAGC-3′) (New England Biolabs, Ipswich, MA, USA), which creates a three nucleotide overhang complementary to our 3WJ after cutting. The DNA substrates were concentrated from the PCR product and purified using gel electrophoresis. The purified DNA was digested with SapI and ligated with the Y junction overnight at 16 °C. The final product was then purified by gel electrophoresis. A representative gel can be seen in Appendix A. DNA concentration was then determined using NanoDrop Spectrophotometer (ND-1000, Thermo Fisher, Waltham, MA, USA) and stored at 4 °C before being used for experiments.

### 4.4. Assembly of Nucleosome Complexes

Nucleosomes were assembled on the end-labeled DNA substrate using a gradient dialysis method optimized from our previous research [33,34]. Recombinant human histone octamers were purchased from The Histone Source (Fort Collins, CO, USA) for use in assembly. Before assembly, histones were dialyzed against initial dialysis buffer (10 mM Tris pH 7.5, 2M NaCl, 1mM EDTA, 2mM DTT) at 4 °C for 1 h. End-labelled DNA was then added to the octamer solution at a 2:1 octamer: DNA ratio and the initial dialysis buffer was replaced with low salt buffer (10 mM Tris pH 7.5, 2.5 mM NaCl, 1mM EDTA, 2mM DTT) over 24 h using a dialysis pump. The nucleosomes were then dialyzed for 1 h against a fresh low salt buffer before being diluted to 300 nM and stored at 4 °C.

### 4.5. Atomic Force Microscopy Imaging of Dry Samples in Air

Freshly cleaved mica was functionalized with a 167 μM solution of 1-(3-aminopropyl)-silatrane (APS) for 30 min at room temperature, rinsed, and dried with a gentle flow of argon as described in [35]. Samples were diluted to 2 nM concentration in imaging buffer (10 mM HEPES pH 7.5, 4 mM MgCl_2_), deposited on the mica surface for 2 min, rinsed with water, and dried with gentle argon flow. Samples were stored in vacuum and argon until ready for imaging on a Nanoscope V system using TESPA probes (Irvine, CA, USA). A typical image captured was 1 μm × 1 μm in size with 2 nm/pixel resolution.

### 4.6. High-Speed Atomic Force Microscopy Imaging in Aqueous Buffer

High-speed AFM imaging was performed as described in our previous literature [28,33,36]. Briefly, a thin piece of mica was punched into 2 mm diameter circular pieces, which were glued onto the sample stage of the HS-AFM (RIBM, Tsukuba, Japan). For functionalization of this mica surface, 2.5 μL of 500 μM APS solution was deposited onto the mica and incubated for 30 min by covering with a wet cap. The mica surface then was rinsed with 20 μL of deionized water. Then, 2.5 μL of the DNA or nucleosome sample was deposited onto the APS functionalized mica surface and incubated for 2 min. The sample was then rinsed and put into the fluid cell containing the imaging buffer described above. During the whole process, the mica surface was not allowed to become dry. Imaging was carried out by HS-AFM using electron beam deposition (EBD) tips. The typical scan size was 300 × 300 nm with a scan rate of 600 ms/frame.

### 4.7. Image Analysis

DNA contour length analysis of naked DNA was performed by tracing the DNA from the free end to the center of the 3WJ. The junction arm length analysis was performed by measuring from the free end of the 3WJ to the joint. Flank measurements for nucleosomes were carried out as described previously [34]. Briefly, the lengths were obtained by measuring from the end of the 3WJ to the center of the nucleosome for the labeled arm and measuring from the center of the nucleosome to the free end of DNA for the non-labeled arm. Each flank length has 5 nm subtracted from the total to account for the size contributed by the histone core. These measurements were conducted using Femtoscan software (Advance Technologies Center, Moscow, Russia). The mapping of the nucleosomes was completed in Microsoft Excel. The histograms were generated using Origin software (OriginLab Corporation, Northampton, MA, USA). Nucleosome height was measured using cross-section analysis in Femtoscan, with the peak height being recorded. A single cross section was drawn across the center of the nucleosome for each frame measured.

## Figures and Tables

**Figure 1 ijms-23-11404-f001:**
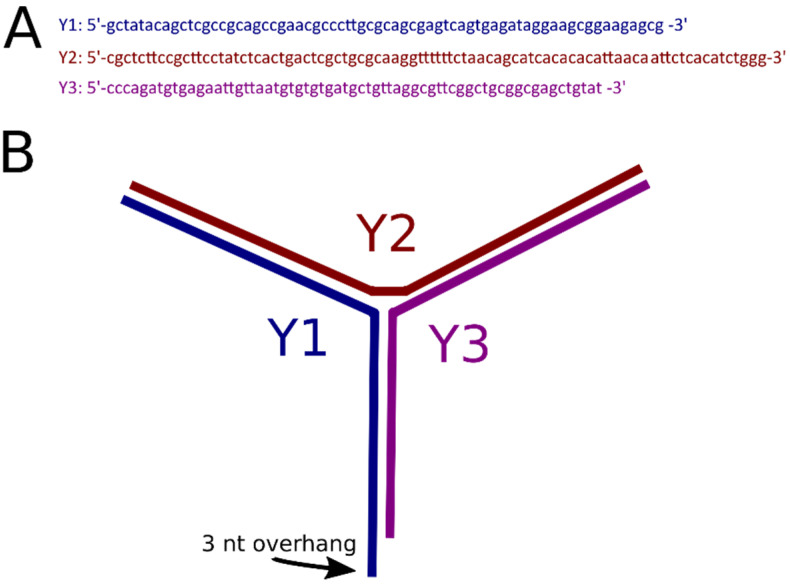
3WJ assembly. (**A**) Oligonucleotides used in 3WJ assembly. (**B**) Schematic of 3WJ. The arm lengths are 40 bp, 39 bp, and 20 bp between Y1–Y2, Y2–Y3, and Y1–Y3, respectively. The 3 nt overhang is GCT. The addition of 6 T’s in the center of the Y2 ssDNA and at the 3WJ intersection site was to help the DNA flanks to prefer the T and Y conformations. These T’s do not have a complementary DNA.

**Figure 2 ijms-23-11404-f002:**
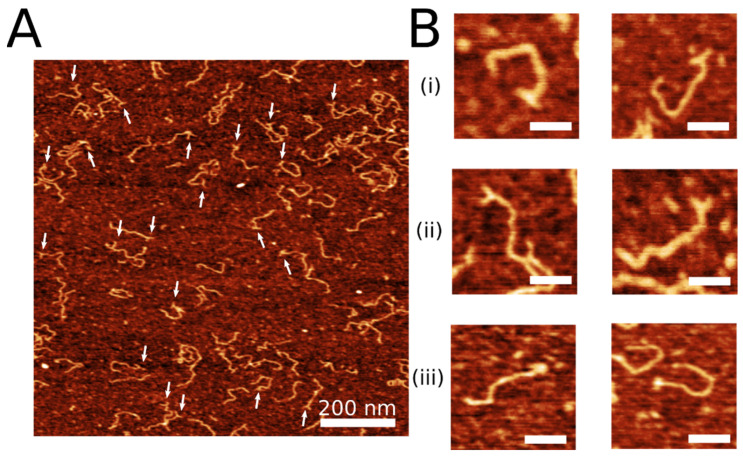
3WJ label on free DNA. (**A**) Representative AFM image of 3WJ label on DNA. White arrows point to label. Scale bar is 200 nm. (**B**) Snapshots of DNA with open “T” label conformation (**i**), bent “Y” label conformation (**ii**), and condensed “bulge” label conformation (**iii**). The AFM images are taken in air. Scale bars indicate 50 nm.

**Figure 3 ijms-23-11404-f003:**
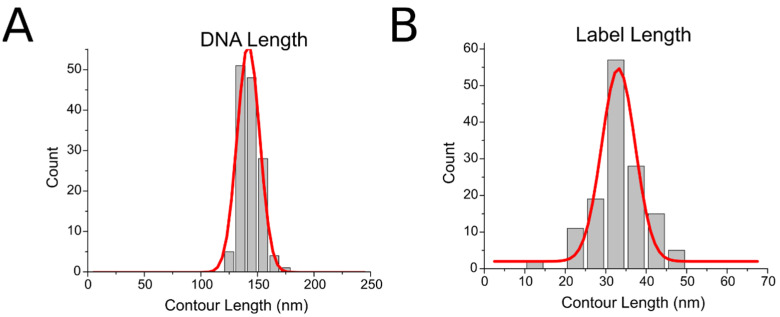
Contour length measurements of (**A**) DNA contour length and (**B**) Lengths of the arms. DNA length is 142 ± 10 nm (S.D.) and label length is 33 ± 4 nm (S.D.).

**Figure 4 ijms-23-11404-f004:**
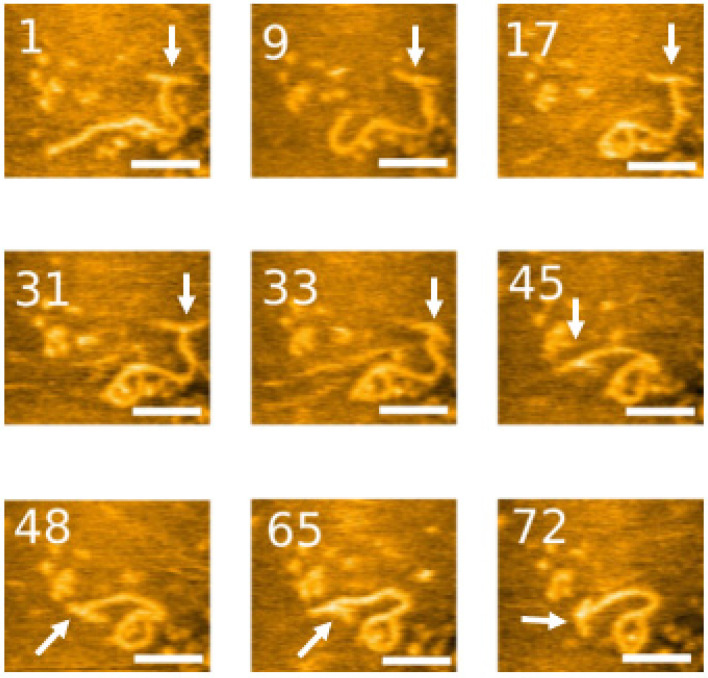
High-speed AFM imaging of labeled DNA substrate. 3WJs are indicated with arrows. Scale bar indicates 40 nm. The numbers in each image are their frame number in the movie file. The complete set of frames can be visualized in Appendix A. The numbers on the snapshots indicate the frame number. A frame is taken every 600 ms.

**Figure 5 ijms-23-11404-f005:**
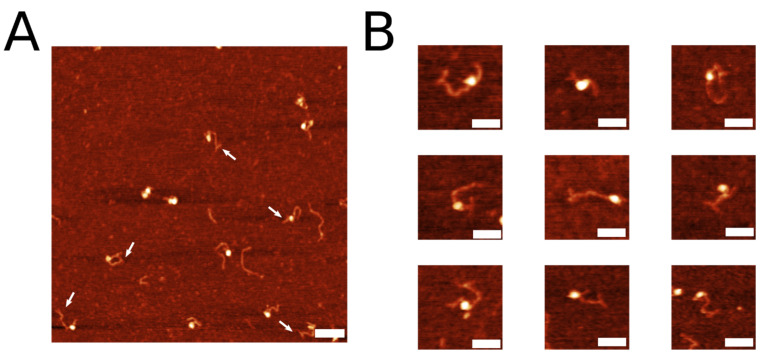
AFM images of nucleosomes assembled on labeled DNA. (**A**) A representative frame with images of assembled nucleosomes. White arrows indicate 3WJ labels. Scale bar indicates 100 nm. (**B**) Selected snapshots of nucleosomes with label present. Scale bar indicates 50 nm.

**Figure 6 ijms-23-11404-f006:**
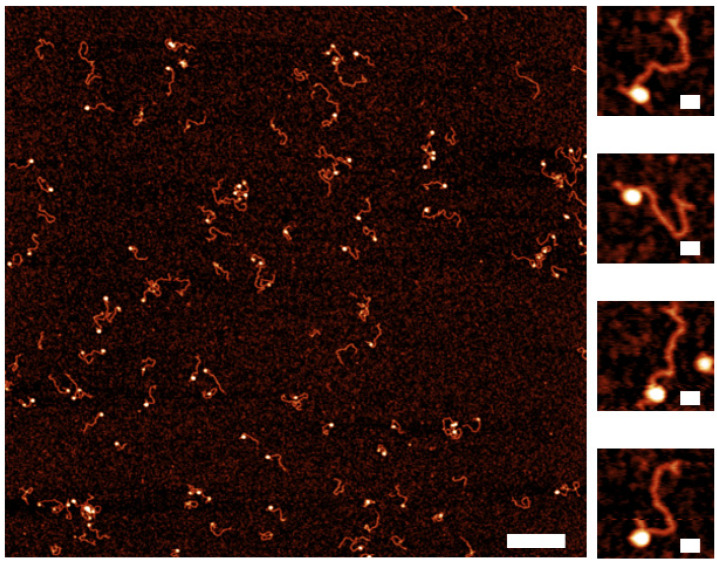
AFM images of dried nucleosome sample assembled on a dinucleosome construct containing the Widom 601 sequence. The snapshots display nucleosomes bound to the Widom 601 sequence and a clear 3WJ. The scale bars are 300 nm and 25 nm for the larger image and snapshots, respectively.

**Figure 7 ijms-23-11404-f007:**
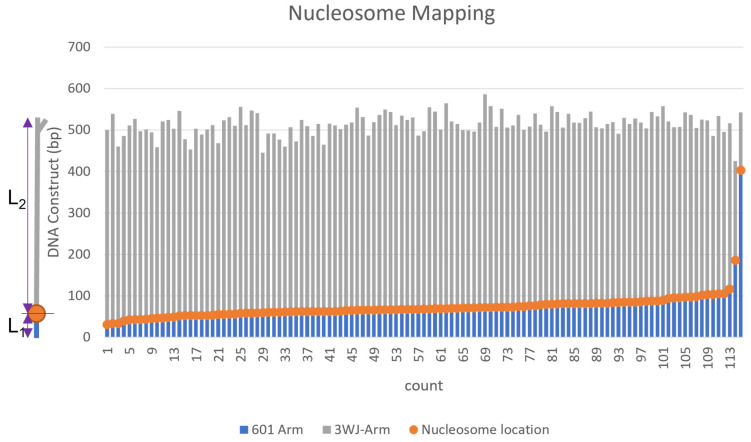
Mapping data from the analysis of nucleosomes assembled on the DNA template containing the nucleosome-specific and non-specific sequences. On the left, is a schematic of the DNA construct with the nucleosome bound (orange dot). The nucleosome position was measured from the center of the nucleosome particle to the DNA end. The grey bars (L_2_) correspond to the distance measured from the end of the 3WJ to the center of the nucleosome, and the blue bars (L_1_) show the distance from the center of the nucleosome to the unlabeled end of the DNA. 5 nm was subtracted from the measurement of each DNA flank to account for the size contributed by the nucleosome core. The variable overall contour lengths of the DNA are due to the varying wrapping efficiency of the nucleosome.

**Figure 8 ijms-23-11404-f008:**
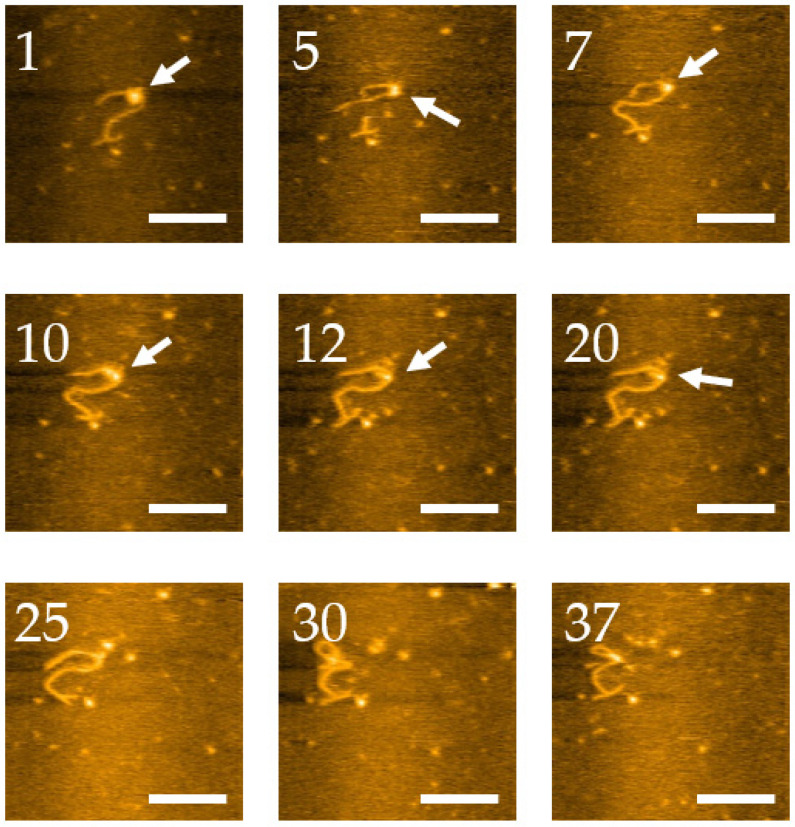
High-speed AFM imaging of nucleosome dynamics on labeled DNA. The nucleosome is indicated with arrows. Scale bars indicate 50 nm. The numbers in each image are their frame number in the movie. The numbers in each image are their frame number in the movie. The complete set of frames can be visualized in Appendix A.

**Figure 9 ijms-23-11404-f009:**
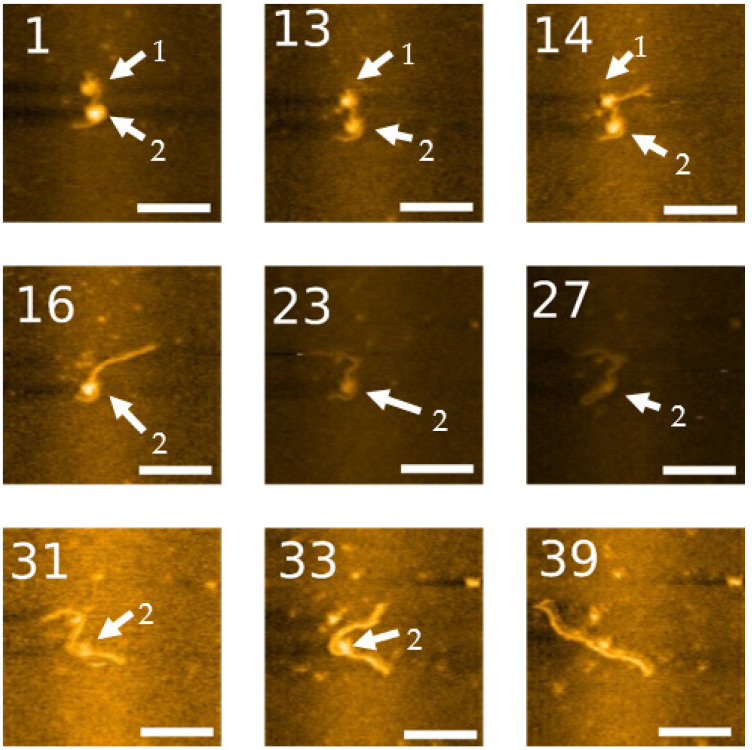
High-speed AFM imaging of dinucleosome dynamics on labeled DNA. The arrow labeled “1” points to a nucleosome assembled near the 3WJ, which can be easily visualized in frame 14. A second nucleosome is marked with an arrow labeled “2”, which remains assembled longer than the other nucleosome. Scale bar indicates 50 nm. The numbers in each image are their frame number in the movie. The complete set of frames can be visualized in Appendix A.

**Table 1 ijms-23-11404-t001:** The yield of different arrangements of the arms of the junction label.

Label Conformation	Frequency
“T” Label	52.6% (n = 72)
“Y” Label	39.4% (n = 54)
“Bulge” Label	8.0% (n = 11)

## Data Availability

All data are included in this paper.

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
