# Peer review of "Three-Way DNA Junction as an End Label for DNA in Atomic Force Microscopy Studies"

_ijms, 2022, doi:10.3390/ijms231911404_

Round 1
Reviewer 1 Report
This methodological advance for end labelling of DNA as fiducial markers for AFM studies of DNA-protein interactions builds on the previous work of the Thomson group. Rather than use a single stranded DNA loop, they design and incorporate a double-stranded 3-way T junction (3WJ).
It is very well written and presented manuscript that clearly demonstrates the viability and utility of the 3-way junction T end label. The authors show this by imaging assembly/disassembly of nucleosomes at specific sites on designed linear DNA templates including the end label. The experiments are conducted with rigour and the quality of the imaging and data analysis highlight the applicability of this new type of end label.
There are some points the authors should consider to further improve the manuscript;
1) The authors cite one paper of the Thomson group in Nucleic Acids Research (Ref 20) but should also include Methods 60 (2013) 122-130, as this gives more details on end-labelling DNA templates with ssDNA loops and is an update on the method using PCR amplification on which this current manuscript approach is based.
2) At lines 56 – 59, the authors query two points about the method developed by the Thomson group. Firstly, they incorrectly state that the yield for the ssDNA labels is low. It is correct that in the original method published in NAR 40 13 e99 that the yield was ~50%, however the updated method in Methods 60 (2013) 122-130, which is exactly comparable with the current work at 70% and why this other paper should be cited and discussed in this context.
3) Secondly, they query that the flexibility of the ssDNA loops leads to low contrast in the AFM imaging. While they have a point, this very much depends on the imaging conditions, such as surface binding mechanism and whether or not studies are conducted in air or liquid. The T end label is clearly an important addition to the arsenal of fiducial DNA end labels, but they are also quite dynamic showing at least 3 different conformations and in the HS-AFM timelapses not always distinguishable. This statement should be modified to take account of the fact that the use of either of these end labels will depend on the application and imaging modalities.
4) The sequence design of the T label is not explained sufficiently clearly. Specifically, does the 6 T run at the junction on strand Y2, have any complementary base-pairing? Was is necessary to introduce a single-stranded part of the T at the junction to ensure it self-assembled reliably? Would this make it more flexible? Is this what causes some % of the 3WJ to exhibit a “bulge”? These issues could be highlighted more clearly in Fig. 1 and in the discussion, as it’s important information.
5) Are images in Figure 2 taken in air ? Please clarify at least in the figure legend.
6) Fig. 3 and discussion: What is the expected contour length of this DNA template, based on B-form DNA and does it correspond well to the measured value.
7) Fig. 4 – what are the frame times in this imaging series? I assume the numbers are frame numbers but there’s no indication of timescale (Note: I did not have access to the supp info for this review assignment).
8) Lines 186-189: The authors discuss that the 3WJ allows reliable determination of the nucleosome binding sequence. This is demonstrated on a carefully designed template, but can they discuss what kind of resolution they have in either terms of bp or nm, to quantify movements of the nucleosomes on the DNA templates.
9) Fig. 8, Frame 10 – would benefit from additional analysis/presentation, e.g. cross-sections, to show that there are really two nucleosomes unbinding as it’s not that clear from the false-colour image alone.
10) Fig. 9: In this series, the T label is sometimes too mobile to be sensed by the AFM tip in some frames. Can the authors comment, whether or not this is an issue for this kind of application i.e. determining protein dynamics on DNA in situ using HS-AFM under liquid.
11) Lines 276-278: Not clear on first reading that this refers to the T design – refer back to a modified Fig. 1.

Author Response
This methodological advance for end labelling of DNA as fiducial markers for AFM studies of DNA-protein interactions builds on the previous work of the Thomson group. Rather than use a single stranded DNA loop, they design and incorporate a double-stranded 3-way T junction (3WJ).
It is very well written and presented manuscript that clearly demonstrates the viability and utility of the 3-way junction T end label. The authors show this by imaging assembly/disassembly of nucleosomes at specific sites on designed linear DNA templates including the end label. The experiments are conducted with rigour and the quality of the imaging and data analysis highlight the applicability of this new type of end label.
There are some points the authors should consider to further improve the manuscript;
1) The authors cite one paper of the Thomson group in Nucleic Acids Research (Ref 20) but should also include Methods 60 (2013) 122-130, as this gives more details on end-labelling DNA templates with ssDNA loops and is an update on the method using PCR amplification on which this current manuscript approach is based.
Response: We thank the reviewer for their comments. We included the new citation to the Thomson group– reference 21.
2) At lines 56 – 59, the authors query two points about the method developed by the Thomson group. Firstly, they incorrectly state that the yield for the ssDNA labels is low. It is correct that in the original method published in NAR 40 13 e99 that the yield was ~50%, however the updated method in Methods 60 (2013) 122-130, which is exactly comparable with the current work at 70% and why this other paper should be cited and discussed in this context.
Response: We included that the updated methods increased the yield of ssDNA loops. Later experiments increased the yield of the terminal ssDNA marker to ~70% [21].
3) Secondly, they query that the flexibility of the ssDNA loops leads to low contrast in the AFM imaging. While they have a point, this very much depends on the imaging conditions, such as surface binding mechanism and whether or not studies are conducted in air or liquid. The T end label is clearly an important addition to the arsenal of fiducial DNA end labels, but they are also quite dynamic showing at least 3 different conformations and in the HS-AFM timelapses not always distinguishable. This statement should be modified to take account of the fact that the use of either of these end labels will depend on the application and imaging modalities.
Response: We modified this section of the paper to emphasize the benefits in the use of 3WJ in such studies as mapping: “the high flexibility of ssDNA and its relatively low contrast in AFM images compared with dsDNA is a complication in the reliable identification of ssDNA loops at the end of DNA duplexes for some studies. Reliable visualization of a terminal label in HS-AFM studies, is particularly important, as multiple frames per second are used to describe the dynamics of a system. The use of a dsDNA-based label could help overcome these complications”.
4) The sequence design of the T label is not explained sufficiently clearly. Specifically, does the 6 T run at the junction on strand Y2, have any complementary base-pairing? Was is necessary to introduce a single-stranded part of the T at the junction to ensure it self-assembled reliably? Would this make it more flexible? Is this what causes some % of the 3WJ to exhibit a “bulge”? These issues could be highlighted more clearly in Fig. 1 and in the discussion, as it’s important information.
Response: We included description in the figure legend that the addition of 6 T’s in the center of the Y2 ssDNA and at the 3WJ intersection site was to help the DNA flanks to have the flexibility to adopt the T and Y conformation more frequently. The (T)6 segment does not have
complementary segments in the construct.
5) Are images in Figure 2 taken in air ? Please clarify at least in the figure legend.
Response: Figure 2 shows topography in air and has been clarified in the figure legend. |
6) Fig. 3 and discussion: What is the expected contour length of this DNA template, based on B-form DNA and does it correspond well to the measured value.
Response: The DNA length is 120.3 ± 9 nm, which for the DNA with 356 bp corresponds to the value 0.338 nm/bp. We added this value to the revision.
7) Fig. 4 – what are the frame times in this imaging series? I assume the numbers are frame numbers but there’s no indication of timescale (Note: I did not have access to the supp info for this review assignment).
Response: The data scan rate for HS AFM was 600 ms per frame
8) Lines 186-189: The authors discuss that the 3WJ allows reliable determination of the nucleosome binding sequence. This is demonstrated on a carefully designed template, but can they discuss what kind of resolution they have in either terms of bp or nm, to quantify movements of the nucleosomes on the DNA templates.
Response: Our purpose is to make a marker to clearly identify a particular end of DNA, and start to measure from this location. Enabling the identification of the region in which the nucleosome is bound to the DNA. The analysis of the nucleosome dynamics using this labeling of DNA is our
current goal. The resolution of the images is determined by the pixel density, these images are scanned at a resolution of 2 nm/pixel, which translates to a resolution of a few bp per pixel.
9) Fig. 8, Frame 10 – would benefit from additional analysis/presentation, e.g. cross-sections, to show that there are really two nucleosomes unbinding as it’s not that clear from the false-colour image alone.
Response: the height measurements were shown in the supplement (Fig. S3). We modified that paragraph to make it clear.
Unwrapping of DNA from the histone core starts after frame 10, with a complete dissociation in frame 25. This conclusion is supported by the height measurements of nucleosomes shown in Fig. S3. The initial height of the nucleosome with the value of 2.5 nm gradually decreases over time to 1 nm (frame 17), after which the core dissociates. Note that the 3WJ label in this frame is still clearly visible with primarily T-shape briefly adopting a “Y” conformation as shown in frame 20.
10) Fig. 9: In this series, the T label is sometimes too mobile to be sensed by the AFM tip in some frames. Can the authors comment, whether or not this is an issue for this kind of application i.e. determining protein dynamics on DNA in situ using HS-AFM under liquid.
Response: The 3WJ is visible through most of the frames, despite its dynamics. The arm can be off the surface becoming non-visible but appears on the following frame allowing for unambiguous identification of the DNA end. Once we know which end is the 3WJ end, we can track it through each frame, even if in some frames the 3WJ is not visible on AFM.
11) Lines 276-278: Not clear on first reading that this refers to the T design – refer back to a modified Fig. 1.
Response: We changed the wording from stacked T-type orientation to T-conformation to make the reading smooth.

Reviewer 2 Report
The paper of Sun et al. deals with the methodology to clearly detect and label proper end of DNA strain. For that, they built special DNA origami construct, and presented its application using High-Speed AFM.
The study is clearly designed and presented. I would like only to have clear information about color scale used for presented AFM images (I guess, this are topography images, so the color scale refer to height).
Can we get more details about probes used in the AFM experiments? There are several versions of TESPA probes. Are the EBD tips home-made? If so, could you provide some characteristics for them?
This is are, however, minor problems. The only reason I'm actually backing for major review, is missing of supplementary information. From reviewers duty, I would like to see it before accepting a paper.
Author Response
file is attached

Round 2
Reviewer 2 Report
I'm satisfied with changes made by Authors.